# Compensatory Effect of the *ScGrf3-2R* Gene in Semi-Dwarf Spring Triticale (x *Triticosecale* Wittmack)

**DOI:** 10.3390/plants11223032

**Published:** 2022-11-09

**Authors:** Anastasiya G. Chernook, Mikhail S. Bazhenov, Pavel Yu. Kroupin, Aleksey S. Ermolaev, Aleksandra Yu. Kroupina, Milena Vukovic, Sergey M. Avdeev, Gennady I. Karlov, Mikhail G. Divashuk

**Affiliations:** 1All-Russia Research Institute of Agricultural Biotechnology, 127550 Moscow, Russia; 2Moscow Timiryazev Agricultural Academy, Russian State Agrarian University, 127434 Moscow, Russia

**Keywords:** triticale, dwarfing genes, *Ddw1*, *ScGrf3-2R*, molecular marker

## Abstract

The dwarfness in many triticale cultivars is provided by the dominant *Ddw1* (*Dominant dwarf 1*) allele found in rye. However, along with conferring semi-dwarf phenotype to improve resistance to lodging, this gene also reduces grain size and weight and delays heading and flowering. *Grf* (*Growth-regulating factors*) genes are plant-specific transcription factors that regulate plant growth, including stem growth, in terms of length and thickness, and leaf and fruit size. In this work, we partially sequenced the rye gene *ScGrf3* on chromosome 2R homologous to the wheat *Grf3* gene, and found multiple polymorphisms in intron 3 and exon 4 complying with two alternative alleles (haplotypes *ScGrf3-2Ra* and *ScGrf3-2Rb*). For the identification of these, we developed a codominant PCR marker. Using a new marker, we studied the effect of *ScGrf3-2R* alleles in combination with the *Ddw1* dwarf gene on economically valuable traits in F_4_ and F_5_ recombinant lines of spring triticale from the hybrid combination Valentin 90 x Dublet, grown in the Non-Chernozem zone for 2 years. Allele *ScGrf3-2Ra* was associated with greater thousand-grain weight, higher spike productivity, and earlier heading and flowering, which makes *ScGrf3-2R* a perspective compensator for negative effects of *Ddw1* on these traits and increases prospects for its involvement in breeding semi-dwarf cultivars of triticale.

## 1. Introduction

Triticale (x *Triticosecale* Wittmack) is an artificially created genus of cultivated plants, and is an intergeneric amphidiploid wheat (*Triticum* ssp.) and rye (*Secale* ssp.). Hexaploid triticale (2n = 6x = 42; BBAARR) remains the most commercially demanded type of this species, for which the first cultivars were registered in 1968 [1]. Triticale was developed to combine the high-yield potential and end-product quality of wheat with the adaptivity of rye [2]. Most triticale consumption is as forage and feed grain. At the same time, this natural resource crop has great potential for the use of its processed products in the production of biofuels (ethanol and biogas), the supply and production of the chemicals, paper, construction, and plastics industries, food production (beer and kvass), and bakery and rusk products [3,4,5].

The breeding process has improved the feed qualities of triticale and eliminated undesirable traits such as pre-harvest sprouting. However, lodging has long been a problem of growing this crop. Lodging reduces the efficiency of photosynthesis, promotes pre-harvest sprouting and disease development, slows maturation, and hampers harvesting [6,7]. The solution of the lodging problem was found in the introduction of semi-dwarfing genes into commercial wheat cultivars. Dwarfing genes reduce plant height and increase yield by redistributing plant resources in favor of the developing head [8,9]. However, these genes also tend to reduce the grain weight, impair the absorption of nitrate nitrogen from soil, and reduce the drought resistance [10,11].

Plant height in triticale can be reduced using either wheat or rye dwarfing genes, or both. Dwarfing genes are classified depending on their response to exogenous gibberellic acid (GA): gibberellin-insensitive and gibberellin-sensitive. To date, 27 genes reducing the height of bread wheat stem have been identified [10,12,13]. The gibberellin insensitivity *Rht-B1b* (synonym with *Rht-1*, *Reduced height-1*) gene was the first to be successfully introduced during the “Green Revolution”. It is widely distributed in wheat and triticale varieties worldwide [14]. The decrease in plant height due to *Rht-B1b* varies from 10% to 25% in common wheat, and from 25% to 35% in durum wheat, in comparison to the carriers of the wild-type allele *Rht-B1a* [15,16,17,18,19,20,21]. Semi-dwarf cultivars and breeding lines of wheat carrying *Rht-B1b* show higher yield than wild-type tall plants [15,19,20,21]. Among the 14 rye dwarf genes known today [22], *Ddw1* (*Dominant dwarf 1*) has the highest agronomic value. The dominant allele of this gene reduces the height of diploid and tetraploid rye plants by 40% and 55%, respectively, but at the same time reduces the grain weight and spike productivity, and delays heading and flowering [23,24,25,26,27,28]. *Ddw1* is successfully used in European triticale breeding programs [8,29,30]. In rye, *Ddw1* manifests pleiotropically by shortening the stem internode, and increasing tillering, size of leaves and spikes, number of spikes, and grains per spike [31].

Growth-regulating factors (GRFs) are plant transcription proteins that participate in the regulation of growth and development. The *Grf* gene was first found in rice as *OsGRF1*; it encodes a protein that regulates the response to gibberellin, which promotes stem elongation [32]. Since the discovery of the first *Grf*, they have been reported in various species. In particular, nine members of the *Grf* family have been identified in *Arabidopsis thaliana* [33,34], 14 in maize [35], and 12 in rice [36]. *In Arabidopsis thaliana*, *Grf* knockout mutants are characterized with smaller and narrower leaves compared to wild-type genotypes [33,37,38]. In rice, the suppression of *Grf3*, *Grf4*, and *Grf5* leads to dwarfism and delay in inflorescence development [39], while increased *Grf* expression results in a significant increase in the length of the panicle, and an increase in the length, width, and weight of the caryopsis [40,41,42,43]. *Grf1* overexpression in maize increases the number of dividing cells, resulting in larger leaves, although plant height is reduced [44].

In bread wheat, 30 *TaGRFs* were identified in 12 linkage groups and classified into four phylogenetic groups [45,46]. Previously, we assessed the *TaGrf3-2D* sequence diversity in common wheat (*Triticum aestivum*, BBAADD) and *Ae. tauschii* (DD), and revealed its influence on grain weight and size in wheat [26]. The effect of the allelic state of *TaGrf3-2A* on thousand-grain weight and spike weight in *Ddw1-*carrying triticale was shown, and the possibility of using *TaGrf3-2A* to compensate for the negative effects of *Ddw1* was demonstrated [47]. In addition, the *TaGrf3-2Ab* allele that highly likely originated from bread wheat, Bezostaya 1, was associated with earlier heading and better grain performance in Krasnodar Krai [48].

Thus, recent studies have proved the significance of *TaGrf3* as a potential means of compensating for the negative effects of dwarfism in wheat and triticale. However, the *Grf* genes in rye have not yet received sufficient attention. In this work, we studied the effect of the rye *ScGrf3-2R* gene on important valuable agronomic traits in spring triticale.

## 2. Results

### 2.1. Partial Sequencing of ScGrf3-2R

The search for a gene homologous to wheat’s *TaGrf3-2A* in the rye Lo7 genome sequence led to the discovery of the most similar fragment to it on chromosome 2R, designated *ScGrf3-2R*. Based on the found sequence and sequences of *TaGrf3* wheat gene homologs, we designed primers specific for the rye genome for one of the gene fragments. Using these, we amplified the *ScGrf3-2R* gene fragment in two rye varieties, Novaya Era and Saratovskaya 7, and in two triticale varieties, Khongor and Dublet. Amplicon sequence analysis showed that both rye and triticale have two different alleles of this gene. The allele found in the Lo7 reference rye genome, in addition to the allele present in Saratovskaya 7 and Dublet, was designated *ScGrf3-2Ra*, and the allele found in Khongor was designated *ScGrf3-2Rb*. Novaya Era was found to be heterozygous, and both *ScGrf3-2Ra* and *ScGrf3-2Rb* were found in it.

The sequenced gene fragment covers most of the third intron and the last fourth exon of the gene, including a small part of the 3’UTR. It should be noted that *ScGrf3-2Ra* and *ScGrf3-2Rb* alleles are very different. In the third intron, there are single nucleotide polymorphisms (SNPs) and 10 insertions/deletions (indels) ranging in size from 1 to 28 nucleotides. The coding sequence of the fourth exon contains one double-nucleotide and four single nucleotide polymorphisms, and an insertion of a triplet of nucleotides in *ScGrf3-2Rb*. At the same time, three of the four SNPs in the coding sequence are synonymous, and the remaining polymorphisms lead to changes in the amino acid sequence. The last exon encodes the C-terminal fragment of the protein, which does not contain conserved domains. According to the PROVEAN prediction, the found changes should not significantly affect the functioning of the protein (Table 1).

Therefore, we identified an allelic polymorphism in the 4th exon between the *ScGrf3-2Ra* and *ScGrf3-2Rb* alleles. Even synonymous nucleotide substitutions can affect translation efficiency, since different tRNAs will be required for different alleles [49]. In the case of *ScGrf3-2Ra* and *ScGrf3-2Rb*, this concerns both nonsynonymous (T263S, N359S) and synonymous substitutions, resulting in non-isoaccepting codons (T290, F313). In addition, the found substitutions and insertion G319_F320insG are located at the carboxylic end of the GRF protein, in which the function of transcription activation via protein–protein interaction was revealed [32,33,40]. In the third intron, we found indels ranging in size from 1 to 28 nucleotides, which can affect the efficiency of gene expression and also lead to phenotypic differences [50]. Thus, we can assume that the revealed differences in the nucleotide and deduced amino acid sequence may be functional for the studied GRF protein.

### 2.2. Development of Allelic-Specific PCR Marker for ScGrf3-2R

To determine the allelic state of *ScGrf3-2R* in triticale lines, we developed a codominant Sequence-Tagged Site (STS) marker based on the presence of insertions/deletions in the third intron of the gene (Figure 1). 

Primers were designed using Primer-BLAST (NCBI) and their specificity was preliminary tested based on the alignment between rye and wheat homologous genes. The expected amplicon size is 220 bp for *ScGrf3-2Ra* (Saratovskaya 7, Dublet) and 180 bp for *ScGrf3-2Rb* (Novaya Era, Khongor). Conditions for PCR amplification are shown in the Materials and Methods section. The observed results of testing STS marker on plant material were consistent with those expected (Figure 2). 

Parental triticale varieties of the hybrid combination Valentin 90 x Dublet differ in the allelic state of *ScGrf3-2R* and the *Ddw1* dwarf gene. The spring variety of triticale Dublet carries *ddw1* (wild type, tall plant) and *ScGrf3-2Ra*, whereas the winter variety Valentin 90 has *Ddw1* (dwarfing allele, short plant) and *ScGrf3-2Rb*. All recombinant lines derived from F_3_ plants used in this study were selected for a spring habit. For genotyping recombinant lines for *ScGrf3-2R*, we used the STS marker developed in this study, and for genotyping for *Ddw1*, the REMS1218 microsatellite marker was applied.

### 2.3. Association between Genotypes and Valuable Agronomic Traits

To assess association between the allelic state of *Ddw1* and *ScGrf3-2R*, on one hand, and the studied plant traits, on the other hand, we studied individual effects of *Ddw1* and *ScGrf3-2R* (Appendix A) and their complex interaction (Appendix A).

#### 2.3.1. Individual Effects of Ddw1 and ScGrf3-2R

The dominant dwarfing gene *Ddw1* significantly reduced the height of spring triticale plants by 28.3 cm (31%) and 27.6 cm (29.7%) in 2018 and 2019, correspondingly, confirming the data of our previous studies. Additionally, a significantly lower productivity of the main spike was observed in the *Ddw1* carriers in comparison to the lines with the wild-type allele. The grain weight per spike was lower in short plants than in tall wild-type plants in both years by 0.15 g (6% and 8%, respectively). The decrease in spike productivity was mainly due to a decrease in the thousand-grain weight (TGW): in *Ddw1*-carrying lines, TGW was 4 g (10%) and 3.8 g (7%) lower in 2018 and 2019, respectively, compared to tall lines. Due to *Ddw1* the number of grains in the main spike decreased in both years insignificantly, by 1–2 pieces (2–3%) (Appendix A).

In both years of the field experiment, *ScGrf3-2Ra* demonstrated positive effect on the productivity of the main spike. The grain weight in the main spike in the *ScGrf3-2Ra*-carrying lines was 0.17 g (9%) and 0.3 g (12%) higher in 2018 and in 2019, respectively, than in the *ScGrf3-2Rb*-carriers. The increase in productivity occurred both due to an increase in the grain number per spike by 3 pcs (7%) and 5 pcs (9%) in 2018 and 2019, respectively, and due to an increase in the TGW by 1 g (2%) in both years. The allelic state of *ScGrf3-2R* showed significantly affected plant height: triticale lines carrying *ScGrf3-2Ra* were lower by 8 cm (10%) in 2018 and 3.7 cm (5%) in 2019 compared to lines carrying *ScGrf3-2Rb*. Spring triticale plants homozygous for *ScGrf3-2Ra* had more compact spikes compared to *ScGrf3-2Rb* in both years of field trials, which was due to a decrease in the spike length and an increase in the spikelet number per spike (Appendix A).

#### 2.3.2. Interaction of Ddw1 and ScGrf3-2R

Tall plants (*ddw1 ddw1*) carrying *ScGrf3-2Ra* had a significantly higher grain number per spike, by 9 pcs (22%) in 2019 and by 6 pcs (13%) in 2018, compared to tall plants carrying *ScGrf3-2Rb*; among semi-dwarf lines (*Ddw1 Ddw1*), the trend was the same, but the effect was not statistically significant (*p* < 0.05) in both years (Figure 3, Appendix A). 

In tall plants of spring triticale (*ddw1 ddw1*) grown in 2018, *ScGrf3-2Ra* significantly increased the grain weight in spike by 0.5 g (25%); among short plants (*Ddw1 Ddw1*) this effect was not statistically significant (*p* < 0.05), but the trend persisted. In 2019, both tall (*ddw1 ddw1*) and short (*Ddw1 Ddw1*) plants homozygous for *ScGrf3-2Ra* had higher grain weight in spikes, by 0.3 g (11–13%), compared to the *ScGrf3-2Rb* carriers (Appendix A).

In 2018, *ScGrf3-2Ra* increased TGW both in tall (*ddw1 ddw1*) and short (*Ddw1 Ddw1*) plants by 2.4 g (6%) and 2.3 g (6%), respectively. Short plants (*Ddw1 Ddw1*) homozygous for *ScGrf3-2Ra* in 2019 had a significantly higher TGW, by 2.1 g (4%), compared to short plants with the *ScGrf3-2Rb* genotype (Figure 4, Appendix A).

In 2018, in short plants (*Ddw1 Ddw1*), *ScGrf3-2Ra* reduced plant height by 6.1 cm (9%) relative to *ScGrf3-2Rb*; in tall plants (*ddw1 ddw1*), the same tendency was not statistically significant (*p* < 0.05). In 2019, in tall lines (*ddw1 ddw1*), plant height was reduced due to *ScGrf3-2Ra* by 6.8 cm (7%), whereas in short lines the same tendency was observed, although it was not statistically significant (*p* < 0.05). Plants homozygous for both *Ddw1* and *ScGrf3-2Ra* were the shortest, with a mean height of 59.8 cm (2018) and 64.4 cm (2019). The number of internodes in spring triticale plants remained equal to five during two years of the field experiment in all studied genotypes, the decrease in plant height was achieved by uniformly reducing the length of each individual internode, and the interaction between *ScGrf3-2Ra* and *Ddw1* almost always resulted in shorter internodes (Appendix A). In both years of the experiment, plants homozygous for both *ScGrf3-2Ra* and *Ddw1* had the highest harvesting index (Appendix A). Therefore, the evaluation of the *ScGrf3-2R* effects on agronomic traits in spring triticale demonstrated that the *ScGrf3-2Ra* allele exhibits a positive effect on plant productivity, i.e., increases grain number and grain weight per spike, and TGW. Additionally, *ScGrf3-2Ra* showed its ability to mitigate the negative effect of *Ddw1* on productivity traits.

In both the 2018 and 2019 experiments, short plants (*Ddw1*-genotype) headed and flowered 3–4 days and 2–4 days later, correspondingly, than tall plants (*ddw1*-genotype). In short plants (*Ddw1 Ddw1*), *ScGrf3-2Ra* accelerated heading by 7 days in 2018 and by 3 days in 2019, and hastened flowering by 8 days in 2018 and by 3 days in 2019. Among tall plants (*ddw1 ddw1)*, *ScGrf3-2Ra* accelerated flowering and heading in both years, albeit not statistically significant (*p* < 0.05, Appendix A). Therefore, the *ScGrf3-2Ra* allele leads to a reduction in the transition from the vegetative to the generative phase, despite the negative effect of the dominant dwarfism gene *Ddw1*, which delays it.

## 3. Discussion

Dwarfing genes, and *Ddw1* in particular, are known to have pleiotropic effects on plant traits. In our previous works in spring triticale [22,23,27], *Ddw1*, in addition to lowering the height, was shown to affect productivity traits, i.e., reduce grain weight and number per spike, and thousand-grain weight (TGW), and also lead to later heading.

In this study, we evaluated the manifestation of *ScGrf3-2R* and *Ddw1* in spring triticale recombinant F_4_ and F_5_ lines grown in a two-year field experiment in the Non-Chernozem zone, so the effects of gene alleles under different weather conditions were analyzed. In both years, the effect of the *ScGrf3-2R* and *Ddw1* allelic state was stable and unidirectional.

We showed that the allelic state of *ScGrf3-2R* in semi-dwarf plants carrying *Ddw1* affects plant productivity, TGW, and grain weight per spike, demonstrating a partial compensatory effect of *ScGrf3-2R* against *Ddw1*. In our previous study [26], we described the effect of *TaGRF3-2D* allelic state on grain weight and size in the common wheat collection, and this confirms our assumption about similar effects of the *ScGrf3-2R* gene on plant traits. The dwarfing effect of *Ddw1* on plant height was 27–28%, which is generally consistent with previous studies [23,24,25,26,27]. The ability of the *ScGrf3-2Ra* allele to mitigate the negative effect of the *Ddw1* allele on productivity traits (TGW, grain number per spike) is promising for the development of new semi-dwarf varieties of spring triticale with increased productivity.

As a result of our experiment, *ScGrf3-2Ra* demonstrated the ability to reduce plant height in spring triticale. Thus, in the breeding of semi-dwarf *Ddw1*-carrying triticale varieties, which are resistant to lodging and are demanded for intensive cultivation technologies, *ScGrf3-2Ra* not only maintains the dwarf stature, but can also decrease it; however, this supplementary dwarfing effect may be disguised by *Ddw1*.

One of *Ddw1*′s drawbacks, both in rye and triticale, is reported to be the delay in heading and flowering [51]. Accelerated transition from the vegetative to the generative phase may be possible in water-deficit and hot regions to escape summer drought and produce large-filled grain. In addition, the earliness per se helps avoid the rain and fog period at harvest, leading to pre-harvest sprouting. In our study, semi-dwarf plants (*Ddw1 Ddw1*) carrying *ScGrf3-2Ra* headed and flowered earlier than those with *ScGrf3-2Rb*, which suggests the possibility of its use in breeding new varieties of spring triticale.

In the present study, a molecular STS allelic-specific marker was developed that can be applied for selecting for desirable *ScGrf3-2R* alleles in breeding programs of triticale and rye to improve their agronomic valuable traits.

The *ScGrf3-2R* gene is homologous to wheat and rice *Grf3* genes, which are characterized by increased expression in the stem and flowers. The *TaGRF3* gene and its homeologs *TaGRF15* and *TaGRF23* in bread wheat have a higher level of expression in the stem and spike than in other organs [52,53]. In rice, *OsGRF3* represses the promoter of the *KNOX* gene *Oskn2*, and the latter is expressed during shoot, inflorescence, and floral development [39,54]. *OsGRF3* also serves as a transcription factor for the *OsbHLH35* gene, which regulates the development of anthers [55]. In our study, polymorphism in *ScGrf3-2R* was found to be associated with phenotypic differences in the parameters of stem (plant height) and spike (time of heading and flowering, number and weight of grains per spike) in triticale. This suggests that *ScGrf3* may also participate in the regulation of plant height, reproductive organ development, and time of heading and flowering in triticale.

The *Ddw1* and *Rht12* genes are probable orthologues [56,57]; the *GA2oxidase* genes are colocalized with these genes [56,58], and this suggests that their dwarfism probably works in the same way and is associated with the work of this enzyme. Therefore, it would be interesting to compare their effects on triticale traits, although keeping in mind that the experiments in different studies were performed under different conditions and genetic background. In this and our previous studies, plant height was decreased by 30–37% due to *Ddw1* and in the study of Hao et al. [59] by 37.7% due to *Rht12*, which demonstrates their similar effects on plant height. Productive traits in triticale are affected negatively by both genes but less dramatically by *Ddw1* compared to *Rht12*: spikelet number per spike was reduced by 0–2% due to *Ddw1* vs. 12.8% due to *Rht12*, grain number per spike decreased by 0–3% vs. 25.1%, and thousand-grain weight was reduced by 9–14% vs. 14.5%. Moreover, harvest index increased by 5–6% due to *Ddw1* and decreased by 16% due to *Rht12* [24,25,27]. It can be preliminarily concluded that the *Ddw1* gene is more beneficial for breeding of semi-dwarf triticale than *Rht12*, although additional direct comparison studies are required.

The latest data reveal a trend in which the height of triticale plants decreased between 1982 and 2010, and since 2011 there has been an increase in the share of higher genotypes in the cultivar market. This can be explained by an increase in demand for forage triticale varieties with high biomass yield, as plant height is one of the main factors affecting biomass yield [8]. The results of our research demonstrate that incorporation of selection for *ScGrf3-2Ra* in breeding schemes potentially can help to increase the supply of semi-dwarf grain triticale varieties in the current market. The same analysis of breeding trends showed that developmental stages did not change over time in triticale cultivars, but there was a slight trend towards earlier heading in later varieties [8]. The selection for *ScGrf3-2Ra* may help to strengthen this trend and contribute to the development of cultivars with earlier heading.

Therefore, our findings may be useful for the development of more productive earlier semi-dwarf grain triticale varieties. These varieties could help promote this crop to be among the top-four grains, along with wheat, rice, and corn, which will allow us to appreciate the benefits of triticale in terms of sustainable agriculture and functional food.

## 4. Materials and Methods

### 4.1. Plant Material

The varieties of winter rye Novaya Era (N.I. Vavilov All-Russian Institute of Plant Industry, St. Petersburg) and Saratovskaya 7 (Federal Agrarian Scientific Center of the South-East, Saratov), and varieties of hexaploid spring triticale Dublet (Danko Hodowla Roślin, Poland) and Khongor winter triticale (P.P. Lukyanenko National Grain Center, Krasnodar) were used for partial sequencing of the *ScGrf3-2R* gene.

The progenies of the F_3_ plants of the Valentin 90 (*Ddw1 Ddw1 ScGrf3-2Rb ScGrf3-2Rb*) x Dublet (*ddw1 ddw1 ScGrf3-2Ra ScGrf3-2Ra*) intercross were used as material for studying the effect of the *ScGrf3-2R* gene on agronomical traits, in interaction with the *Ddw1* dwarfing gene.

During 2016–2017, generations F_2_ and F_3_ were grown in the field under spring sowing with mass selection of spring forms. In 2018–2019, field experiments of homogeneous spring-type families were carried out. Thus, 121 lines were created and used in this study. 

### 4.2. Field Conditions

The field experiment was carried out in 2018 and 2019 at the field experimental station of Russian State Agrarian University—Moscow Timiryazev Agricultural Academy (55°50′30.8″ N 37°33′24.1″ E, Moscow, Russia). The seeds of the F_4_ and F_5_ generations were sown in the first week of May in 2018 and 2019, respectively, on single-row plots 1 m long with a row spacing of 30 cm. In general, 2019 was more favorable for the spring triticale yield; in July of 2018 there was an increased amount of precipitation compared to 2019, but it was within the climatic norm (according to ten-year data) (Table 2) [60]. An increased level of precipitation during the anthesis stage, which took place in the first week of July for some lines, could reduce the grain set, while during grain filling and ripening stages, it could favor disease development such as root rot, *Septoria*, and *Fusarium* head blight [61].

### 4.3. Analysis of Valuable Agronomic Traits 

The study of agronomical traits was carried out in 10 individual plants from each hybrid line, F_4_ and F_5_. The date of the heading and flowering was determined by the onset of the corresponding stage in at least 80% of the plants in the row. Plant height was measured at the main shoot from the tillering node to the top of the spike except for the awns. The length of the main spike and each internode, the number of internodes, the spikelet number, grain number and grain weight in the main spike, and the stem and spike weight were measured in the main shoot. In the plant as a whole, the number of spikes per plant (productive tillering) was determined, in addition to the weight of grain in the secondary shoots (tillers). The thousand-grain weight (TGW) was determined as the thousand-fold ratio of the grain weight from the main spike to the number of grains from the main spike. Spike compactness was defined as the ten-fold ratio of the total number of spikelets in the main spike to the length of the main spike. The harvest index (HI) was calculated as the ratio of the grain mass per plant to the total mass of the plant at harvest.

A total of 1716 and 1794 individual plants were analyzed in 2018 and 2019, respectively.

### 4.4. DNA Extraction, PCR and Sequencing

Plant genomic DNA was extracted from dried leaves according to the protocol using cetyltrimethylammonium bromide (CTAB) [62].

The expected rye *ScGrf3-2R* gene sequence was found in the Lo7 reference rye genome assembly using BLAST (Nucleotide-Nucleotide BLAST 2.8.1+) [63] and the *TaGRF3-2A* (*TraesCS2A02G435100*) gene sequence from wheat [64]. The boundaries of exons and introns of the gene were predicted using the AUGUSTUS online service (https://bioinf.uni-greifswald.de/augustus/submission.php, accessed on 3 April 2022). [65]. Primers for amplification of large gene fragments were selected using the Primer-BLAST (NCBI) online service. We amplified the region of the 3rd intron and the 4th exon of the *ScGrf3-2R* gene in the studied accessions of triticale and rye using the primers designed (GRF-2R_107L: TTCTGGGTCCATATTTTAGCCCG, GRF-2R_107R: CAGCTCACAGACACGTTTGATC).

The PCR was performed in 25 µL reaction volumes containing 70 mM Tris–HCl buffer (pH 9.3), 16.6 mM (NH_4_)_2_SO_4_, 2.5 mM MgCl_2_, 0.2 mM each dNTP, 30 pM forward and reverse primers (Syntol, Moscow, Russia), 0.04 U/µL LR (long reading) Plus polymerase (Sileks, Moscow, Russia), 0.02 U/µL Taq polymerase (Sileks), and 4 ng/µL DNA template; amplification was performed on a Bio-Rad T100 (Hercules, California, USA,). The PCR conditions were as follows: (1) 95 °C—10 min, (2) 45 cycles 95 °C—30 s, 60 °C—30 s, 72 °C—4 min; (3) final elongation 72 °C—10 min.

PCR products were separated in 1.5% agarose gel with TBE buffer (90 mM Tris-HCl, pH 8.3, 90 mM boric acid, 0.1 mM EDTA), using a GeneRuler 100 bp Plus DNA size standard (Thermo Fisher Scientific, Waltham, Massachusetts, USA) in Bio-Rad Sub-Cell horizontal electrophoresis chamber in conjunction with a Power Pac Basic, Bio-Rad power supply. The gels were stained with ethidium bromide and visualized using the Gel Doc XR+ system (Bio-Rad Laboratories, Hercules, California, USA) under ultraviolet light.

PCR products were sequenced by fragments using the Illumina MiSeq new generation platform at LLC «Genomed» (Moscow, Russia). DNA libraries were prepared using the Swift 2S Turbo DNA Library Kit (Swift Biosciences, USA), with the amplicons labeled with DNA barcodes. Sequence assembly from the studied sequences was carried out as described earlier, combining de novo assembly with alignment of contigs to the reference genome region [66]. A preview of the alignment was performed using the Integrative Genomics Viewer 2.12.2 [67]. The resulting sequences were compared with each other using GeneDoc 2.7 software [68]. The translation of the protein-coding nucleotide sequences into amino acids was carried out using the GeneDoc 2.7 program. The prediction of the functional significance of the detected amino acid substitutions in the protein was carried out using the PROVEAN online service [69]. Conserved domains of the proteins were annotated by searching the Conserved Domain Database (NCBI) [70]. 

### 4.5. Molecular Markers

The allelic state of the *Ddw1* gene was determined using primers for the REMS1218 microsatellite sequence (F: 5’-CGCACAAACAAAAACACGAC-3’, R: 5’-CAAACAAACCCATTGACACG-3’) [71] and subsequent fragment analysis on an Applied Biosystems™ 3130 genetic analyzer (USA). Amplification conditions were as follows: (1) 94 °C—5 min, (2) 35 cycles 94 °C—30 s, 60 °C—30 s, 72 °C—1 min; (3) final elongation 72 °C—5 min.

To determine the haplotypes of the *ScGrf3-2R* gene, we used the STS marker designed by us. PCR with primers SCGRF3-2R-HF: CCTGCTTTAAAATGTGCAGCAAC, SCGRF3-2R-HR: AGACTTGCAGCATAGTGACCAA was carried out in a volume of a 25 µL mixture containing 70 mM Tris-HCl buffer (pH 9.3), 16.6 mM (NH_4_)_2_SO_4_, 2.5 mM MgCl_2_, 0.2 mM each dNTP, 30pM forward and reverse primers (Syntol, Moscow, Russia), 0.05 U/µL Taq stained polymerase (Silex, Moscow, Russia), 4 ng/µL DNA template. PCR conditions were as follows: (1) 95 °C for 10 min, (2) 36 cycles for 95 °C—30 s, 60 °C—30 s, 72 °C—1 min; and (3) final elongation at 72 °C for 10 min; amplification was performed on a Bio-Rad T100 (USA). PCR products were separated in a Bio-Rad Sub-Cell horizontal electrophoresis chamber in conjunction with a Power Pac Basic, Bio-Rad power supply (USA) in a 2% agarose gel with TBE buffer (Tris, boric acid, EDTA) in the presence of the molecular weight marker M-100 (Syntol, Moscow, Russia), stained with ethidium bromide and visualized using the Gel Doc XR+ system (Bio-Rad Laboratories, Hercules, California, USA). The expected sizes of amplicons are 220 and 180 bp.

### 4.6. Statistical Analysis 

Statistical analysis was performed using R 4.1.2 [72]. A two-way ANOVA was performed using the car 3.1-0 package [73]. False Discovery Rate (FDR) *p*-value correction of 0.05 was used to identify false positive significant patterns in ANOVA. Pairwise comparisons were performed using the Tukey criteria from the agricolae 1.3-5 package [74]. Only plants homozygous for both studied genes, *Ddw1* and *ScGrf3-2R*, were analyzed. Pairwise comparison of mean values carried out in one-way analysis between homozygous genotypes *Ddw1, Ddw1* (short plants) and *ddw1, ddw1* (tall plants), *ScGrf3-2Ra, ScGrf3-2Ra* and *ScGrf3-2Rb, ScGrf3-2Rb*, in two-way analysis—between dihomozygous genotypes, *Ddw1 Ddw1 ScGrf3-2Ra ScGrf3-2Ra*, *Ddw1 Ddw1 ScGrf3-2Rb ScGrf3-2Rb*, *ddw1 ddw1 ScGrf3-2Ra ScGrf3-2Ra*, and *ddw1 ddw1 ScGrf3-2Rb ScGrf3-2Rb*. Graphs of agronomical traits depending on the genotype were constructed using the ggplot2 3.3.6 package [75]. The significance of the difference between the means was estimated at the 95% confidence level.

## 5. Conclusions

In our work, we developed and tested a molecular STS allelic-specific marker that enables distinguishing the allelic state of the *ScGrf3-2R* gene. In our two-year field experiments in the Non-Chernozem zone, we showed a statistically significant effect of the *ScGrf3-2R* allelic state on important agronomic valuable traits, such as grain number and weight in main spike, thousand-grain weight, and heading and flowering time in semi-dwarf (*Ddw1 Ddw1*) spring triticale plants. While maintaining the semi-dwarf phenotype, *ScGrf3-2R* can partially compensate for the negative effects of *Ddw1* on yield-related traits.

## Figures and Tables

**Figure 1 plants-11-03032-f001:**
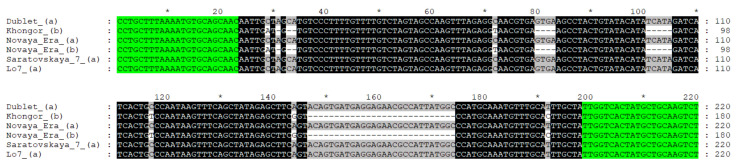
STS marker design based on polymorphisms in the 3rd intron of the *ScGrf3-2R* gene. The sequences of the flowing accessions are shown: triticale, Dublet (*ScGrf3-2Ra*), and Khongor (*ScGrf3-2Rb*); rye, Novaya Era (both alleles), and Saratovskaya 7 (*ScGrf3-2Ra*); reference genome sequence of rye Lo7 (*ScGrf3-2Ra*). The sequences of the designed primers are highlighted in color.

**Figure 2 plants-11-03032-f002:**
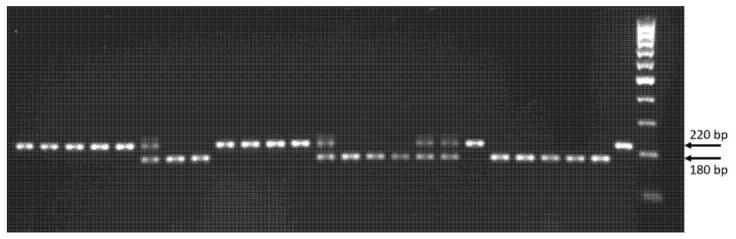
Application of the allelic-specific STS marker for the *ScGrf3-2R* gene. An example of electrophoresis of PCR products amplified from DNA of spring triticale F_3_ plants of the hybrid combination Dublet x Valentin 90. M-100-5 (ZAO Sintol, Moscow, Russia) was used as a molecular weight marker. Expected amplicon sizes: 220 bp for *ScGrf3-2Ra*, 180 bp for *ScGrf3-2Rb*.

**Figure 3 plants-11-03032-f003:**
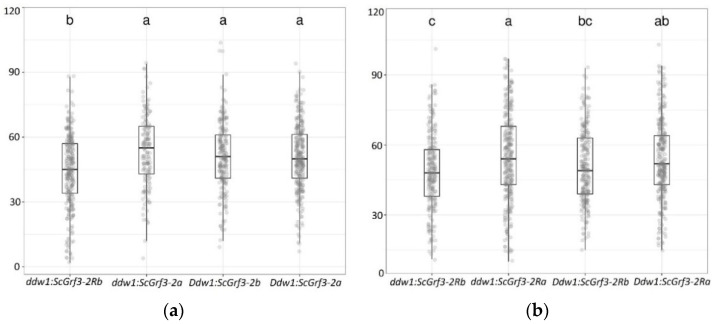
Grain number per main spike (pieces) in recombinant triticale lines with different combinations of *Ddw1* and *ScGrf3-2R* alleles in 2018 (**a**) and 2019 (**b**). Letters denote significantly different groups at a significance level of 0.05 according to Tukey’s test. The rectangles show the interval between the 1st and 3rd quartiles, the vertical lines show the maximum and minimum values, the horizontal line inside the rectangle indicates the median, dots show the trait values of individual plants.

**Figure 4 plants-11-03032-f004:**
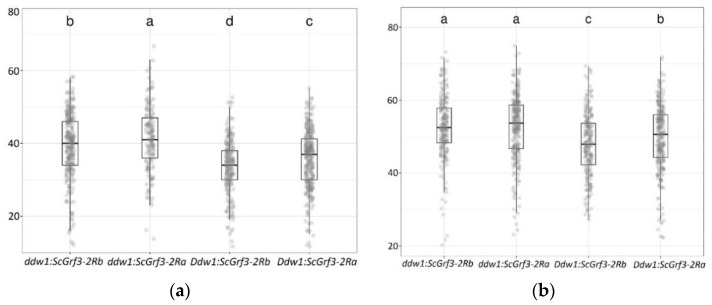
Thousand-grain weight (grams) in recombinant triticale lines with different combinations of *Ddw1* and *ScGrf3-2R* alleles in 2018 (**a**) and 2019 (**b**). Letters denote significantly different groups at a significance level of 0.05 according to Tukey’s test. The rectangles show the interval between the 1st and 3rd quartiles, the vertical lines show the maximum and minimum values, the horizontal line inside the rectangle indicates the median, dots show the individual values of individual plants.

**Table 1 plants-11-03032-t001:** Prediction of the significance of the detected amino acid substitutions for the functionality of the rye GRF3 protein using the PROVEAN online service.

Variant	PROVEAN Score	Prediction (Cutoff = −2.5)
T263S	0.440	Neutral
G319_F320insG	0.079	Neutral
N359S	0.759	Neutral

**Table 2 plants-11-03032-t002:** The weather conditions in Moscow during field experiments.

Month	Amount of Precipitation, mm		Norm, mm	Average Temperature, °C		Norm, °C
	2018	2019	10 Years	2018	2019	10 Years
May	44	58	61	16.1	16.2	13.6
June	54	55	78	17.2	19.6	17.3
July	85	64	84	20.3	16.7	19.7
August	20	48	78	19.8	16.4	17.6

## Data Availability

The data presented in this study are available in Appendix A to this article.

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
