# Peer review of "Compensatory Effect of the ScGrf3-2R Gene in Semi-Dwarf Spring Triticale (x Triticosecale Wittmack)"

_plants, 2022, doi:10.3390/plants11223032_

Round 1

Reviewer 1 Report

Dear sir,

the present paper 'Compensatory Effect of the ScGrf3-2R Gene in Semi-Dwarf 2 Spring Triticale (x Triticosecale Wittmack)' deserves publication in Plants to my opinion. The information presented here is interesting and of applicable importance. However I detected several flaws along the manuscript:

- Intro. Line 35. You must add to 'forage' 'and feed grain'.

- The discussion section must be improved. It is rather short and I don´t understand several sentences in the middle of the last paragraph.

- References. Ref. 57. Diseases of grain crop must be in small case letter. Ref. 39 and 44, the same. Ref. 51, 52, 62. Latin names (Triticosecale, Nicotiana, etc.) in italics.

Best regards

Author Response

Dear Reviewer,

We would like to thank you for the very constructive comments that they will substantially improve the quality of the manuscript. All comments have been taken in consideration and the manuscript was revised according to them. Please find below our point-by-point response to the comments and concerns of the reviewer.

Point 1. Intro. Line 35. You must add to 'forage' 'and feed grain'.

Response 1. An addition has been made.

Point 2. The discussion section must be improved. It is rather short and I don´t understand several sentences in the middle of the last paragraph.

Response 2. We have completed the discussion section and reformulated incomprehensible sentences.

Point 3. References. Ref. 57. Diseases of grain crop must be in small case letter. Ref. 39 and 44, the same. Ref. 51, 52, 62. Latin names (Triticosecale, Nicotiana, etc.) in italics.

Response 3. The bibliography has been revised and corrections have been made.

Kind regards,

Authors

Reviewer 2 Report

In the report “plants-1965769”, the authors found an elite allele ScGrf3-2Ra that overcame the disadvantages of the Ddw1 dwarf gene, including reduced grain size and weight, and late heading and flowering. The findings are meaningful. The paper is well written. The elite allele ScGrf3-2Ra has great potential values in breeding semi-dwarf cultivars of triticale.

Minors:

(1) Line 106: “single nucleotide polymorphisms” is better than “single nucleotide substitutions”.

(2) In figure 2, please remove the dot between b and p.

(3) Table S1-S7 could be combined.

        (4) Table S8-S14 could be combined.

Author Response

Dear reviewer,

We appreciate the positive feedback from you and for your helpful suggestions. We have been able to incorporate changes and reflect all of the suggestions you have provided. Point-by-point responses to comments are given below.

Point 1. Line 106: “single nucleotide polymorphisms” is better than “single nucleotide substitutions”.

Response 1. The phrase has been replaced.

Point 2.  In figure 2, please remove the dot between b and p.

Response 2. Figure 2 has been corrected.

Point 3. Table S1-S7 could be combined.

Response 3. Supplementary tables S1-S7 are merged into S1.

Point 4. Table S8-S14 could be combined.

Response 4. Supplementary tables S8-S14 are merged into S2.

Kind regards,

Authors